# RGBD-GAN: Unsupervised 3D Representation Learning From Natural Image Datasets via RGBD Image Synthesis

**Atsuhiro Noguchi[1] & Tatsuya Harada[1,2]**
[1]The University of Tokyo, [2]RIKEN
{noguchi, harada}@mi.t.u-tokyo.ac.jp

## Abstract

Understanding three-dimensional (3D) geometries from two-dimensional (2D) images without any labeled information is promising for understanding the real world without incurring annotation cost. We herein propose a novel generative model, RGBD-GAN, which achieves unsupervised 3D representation learning from 2D images. The proposed method enables camera parameter–conditional image generation and depth image generation without any 3D annotations, such as camera poses or depth. We use an explicit 3D consistency loss for two RGBD images generated from different camera parameters, in addition to the ordinal GAN objective. The loss is simple yet effective for any type of image generator such as DCGAN and StyleGAN to be conditioned on camera parameters. Through experiments, we demonstrated that the proposed method could learn 3D representations from 2D images with various generator architectures.

## 1 Introduction

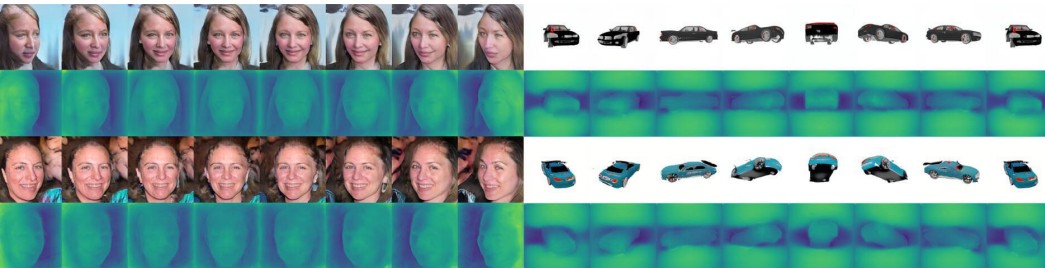

Figure 1: Generated face images from PGGAN and car images from StyleGAN. Images in even rows are generated depth images with colormaps. Though the models are trained on **unlabeled RGB** image datasets, they achieve RGBD image generation as well as explicit control over the camera poses.

Understanding three-dimensional (3D) geometries from two-dimensional (2D) images is important in computer vision. An image of real-world objects comprises two independent components: object identity and camera pose. Object identity represents the shape and texture of an object, and camera pose comprises camera rotation, translation, and intrinsics such as focal length. Learning the representation of these two components independently facilitates in understanding the real 3D world. For example, camera pose invariant feature extraction can facilitate object identification problems, and camera pose variant feature representations are beneficial for the pose estimation of the objects. These tasks are easy for humans but difficult for machines.

Recently, 3D representation learning through 3D object generation has been actively researched. Many techniques are available for learning the relationship between 2D images and 3D objects. Typically used 3D representations are voxel grids (Yan et al., 2016; Wu et al., 2016; Choy et al., 2016; Henzler et al., 2019), point clouds (Fan et al., 2017), and meshes (Rezende et al., 2016; Kato et al., 2018; Wang et al., 2018; Kato & Harada, 2019). For most of the methods, 3D annotations such

as ground truth 3D models (Choy et al., 2016; Fan et al., 2017; Wang et al., 2018), multiple-view images (Yan et al., 2016), or silhouette annotations of objects (Yan et al., 2016; Kato et al., 2018; Kato & Harada, 2019) must be used to reconstruct 3D shape from 2D images. Although these methods achieve 3D object generation by controlling the object identity and camera poses independently, the construction of such datasets requires considerable time and effort. Therefore, a method that can learn 3D representations without any labeled information must be developed. Though some research tackles this problem, their performance is limited when applied to natural images. Rezende et al. (2016) proposed unsupervised single-view 3D mesh reconstruction, but it can only be applied to a primitive dataset. Henzler et al. (2019) recently proposed unsupervised single-view voxel reconstruction on natural images, but the resolution is limited due to the memory constraint.

To realize unsupervised 3D object generation, we employ a different approach, i.e., RGB–Depth (RGBD) image generation. RGBD images comprise the color and depth information of each pixel. The proposed RGBD image generation can be achieved through a simple extension of recently developed image generation models. We propose RGBD Generative Adversarial Networks (RGBD-GAN), which learns to generate RGBD images from natural RGB image datasets without the need of any annotations, such as camera pose and depth annotations, multiple viewpoints for the single objects. The proposed model uses an explicit 3D consistency loss for the generated images; the model generates two RGBD images with different camera parameters and learns them to be consistent with the 3D world. This training pipeline is simple yet effective for generating depth images without supervision and for disentangling a camera pose from the image content. Because the proposed model does not restrict the generator architecture, we can condition any type of image generator (e.g., PG-GAN (Karras et al., 2018), StyleGAN (Karras et al., 2019)) on camera parameters. Figure 1 shows the generation results from the proposed models. As such, our model can generate RGBD images from arbitrary viewpoints without any supervision, and therefore we regard this as "unsupervised 3D representation learning" though a single output cannot represent a full 3D scene.

Our contributions are as follows.

- We propose a new image generation technique, i.e., RGBD image generation, which can be achieved from RGB images without any labeled information such as annotations of camera parameters, depth, or multiple viewpoints for single objects.

- The proposed method can disentangle camera parameters from the image content without any supervision.

- Our method can be used to condition any type of generator on camera parameters because the proposed loss function does not restrict the generator architecture.

## 2 RELATED WORKS

Recently, image generation models have shown significant progress, especially generative adversarial networks (GANs) (Goodfellow et al., 2014). GAN trains a discriminator that estimates the distribution distance between generated and real images; additionally it trains a generator that minimizes the estimated distance. As such, the distribution of training images can be estimated precisely without supervision. Recent interest in generative models pertain to their training stability (Arjovsky et al., 2017; Gulrajani et al., 2017; Miyato et al., 2018) and improvement in quality and diversity (Karras et al., 2018; Brock et al., 2019; Karras et al., 2019). Furthermore, methods to learn 3D morphable generative models from 2D images have been proposed (Tran et al., 2017; Shen et al., 2018; Sitzmann et al., 2019; Nguyen-Phuoc et al., 2019). Tran et al. (2017) and Shen et al. (2018) learned to generate images by controlling camera poses using camera pose annotations or images captured from multiple viewpoints. Although these methods can successfully control an object pose, the scalability is limited owing to the annotation costs. Nguyen-Phuoc et al. (2019) recently proposed a method to disentangle object identity and camera poses without any annotations. This method uses latent 3D features and learns to generate images from the feature projected from the 3D feature with rigid-body transformations. That is, this method uses strong inductive biases regarding the 3D world to learn the relationship between camera poses and images. These image generation models cannot output explicit 3D representations, thus limiting the comprehensibility of the output. Sitzmann et al. (2019) achieved RGB and depth image synthesis from 2D image datasets by unsupervisingly learning occlusion aware projection from 3D latent feature to 2D. The model, how-

ever, requires multiple viewpoints for a single object and camera pose annotations, thus limiting the scalability.

Similarly, Rajeswar et al. (2019) proposed depth image generator trained on unlabeled RGB images. The model can control the pose of generated images without supervision. However, it only works on a synthetic dataset, where the surface-normal of each pixel is easily estimated by the color and location.

## 3 METHOD

In this study, unsupervised 3D representation learning is achieved via RGBD image synthesis. In this section, we first describe the motivation to use RGBD representation in Section 3.1 and we provide the details of our method in Section 3.2.

### 3.1 MOTIVATION

A goal of this research is to construct a model that can generate images $I$ conditioned on camera parameters $c$. However, it is impossible to perfectly model the relationship between $c$ and $I$ without any annotations. Therefore, we alleviate the problem by considering optical flow consistency. Although optical flow is typically used for two different frames in a movie, we used it for images captured with different camera parameters. Optical flow consistency is expressed as the pixel movement between two images.

$$I(x, y, c) = I(x + \Delta x, y + \Delta y, c + \Delta c) \qquad \text{for} \quad \forall x, y, c \tag{1}$$

Here, $x$ and $y$ are pixel coordinates in the image. Considering a small $\Delta c$, this equation can be written as the following partial differential equation.

$$\frac{\partial I}{\partial x}\frac{\mathrm{d}x}{\mathrm{d}c} + \frac{\partial I}{\partial y}\frac{\mathrm{d}y}{\mathrm{d}c} + \frac{\partial I}{\partial c} = 0 \qquad \text{for} \quad \forall x, y, c \tag{2}$$

$\frac{\partial I}{\partial x}$ and $\frac{\partial I}{\partial y}$ can be estimated using ordinary image generation models. Therefore, if $\frac{\mathrm{d}x}{\mathrm{d}c}$ and $\frac{\mathrm{d}y}{\mathrm{d}c}$ are known, then $\frac{\partial I}{\partial c}$ can be calculated. This term can be helpful for conditioning the generator on the camera parameters when optimizing the GAN objective. As $\frac{\mathrm{d}x}{\mathrm{d}c}$ and $\frac{\mathrm{d}y}{\mathrm{d}c}$ remain unknown, we consider a geometric constraint on a homogeneous coordinate. Let $\mathcal{D}$ be the depth, $p = (x, y, 1)$ the homogeneous coordinate of the pixel, $p_{world}$ the world coordinate of the pixel, $R$ the rotation matrix, $t$ the translation vector, and $K$ the camera intrinsics. The camera parameters $c$ are represented herein as $\{K, R, t\}$. $p_{world}$ is constant to $c$. Then, we can calculate the position on an image and the depth from the world coordinate $p_{world}$.

$$\mathcal{D}p = KR\,p_{world} + Kt \tag{3}$$

This facilitates in calculating $\frac{\mathrm{d}x}{\mathrm{d}c}$ and $\frac{\mathrm{d}y}{\mathrm{d}c}$ by estimating the depth $\mathcal{D}$. Hence, we used the RGBD representation for camera parameter conditioning. For depth image $\mathcal{D}$, an optical flow consistency as an RGB image exists, considering the camera parameter change. This facilitates in estimating the depth image $\mathcal{D}$.

$$\mathcal{D}(x, y, c) = \mathcal{D}(x + \Delta x, y + \Delta y, c + \Delta c) + \Delta \mathcal{D} \qquad \text{for} \quad \forall x, y, c \tag{4}$$

Here, $\Delta \mathcal{D}$ can be calculated from Equation 3.

Briefly, training a GAN with the constraints in Equation 1, 3, and 4 is beneficial for learning $\frac{\partial I}{\partial c}$, which benefits camera parameter–conditional synthesis. Additionally, learning a camera parameter–conditional image generation model facilitates in learning depth distributions with the constraint from Equation 1 and 3. The details for each module are explained below.

### 3.2 PROPOSED PIPELINE

The proposed model comprises three components: an RGBD image generator conditioned on camera parameters, RGB image discriminator for adversarial training, and self-supervised RGBD consistency loss. The overview of the pipeline is shown in Figure 2.

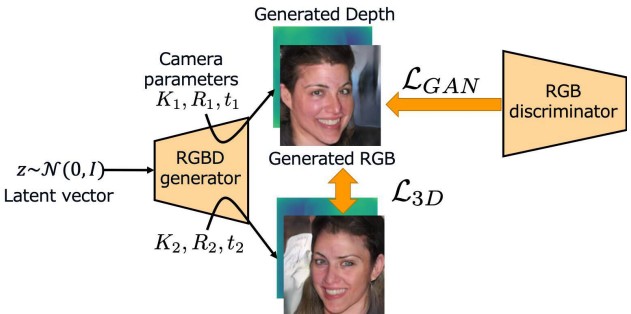

Figure 2: Proposed pipeline. We train the RGBD image generator with the self-supervised 3D consistency loss and adversarial loss for RGB channels. The model generates two RGBD images with different camera parameters and learns them to be consistent with the 3D world.

### 3.2.1 RGBD IMAGE GENERATOR

Considering the success in image generation, the generator of a GAN can estimate complicated distributions. Therefore, we used ordinary RGB image generators such as DCGAN (Radford et al., 2016) or StyleGAN for RGBD synthesis. RGBD synthesis is achieved by adding one channel to the final layer of the RGB generator. Moreover, as described in the experimental section, we can use image generation models through 3D latent representations such as HoloGAN (Nguyen-Phuoc et al., 2019) or DeepVoxels (Sitzmann et al., 2019), which models the 3D world more naturally.

In the proposed pipeline, the generator is conditioned on camera parameters and trained with gradient descent to minimize the self-supervised consistency loss and the adversarial loss described bellow. Because no constraint exists for the generator architecture, any type of generator architecture can be used for RGBD image synthesis, thus resulting in the high applicability of our method.

### 3.2.2 SELF-SUPERVISED RGBD CONSISTENCY LOSS

In Section 3.1, we showed that the optical flow consistency for RGB and depth can facilitate in learning camera parameter–conditional image generation. We approximated the constraint in Equation 1 and 4 by sampling two camera parameters $c_1$ and $c_2$ and minimizing the difference of both sides of the equations for two generated images conditioned on those camera parameters, where $c = c_1$ and $c + \Delta c = c_2$. In this study, the camera parameters are sampled from a predefined distribution $p(c)$ according to the dataset, similarly to HoloGAN. The detailed settings are explained in Section 4.2 and appendix. We limit the maximum values of $\Delta c$ to $30°$ to avoid large occlusion.

The objective function for Equation 1 is similar to the loss used in monocular video depth estimation (Zhou et al., 2017). Using Equation 3, we can calculate the 3D position of each pixel when an RGBD image is viewed from different viewpoints. Therefore, images captured from $c_1$ can be rendered by sampling the pixel values from RGBD images captured from $c_2$. This operation is typically called as "warp" operation and is implemented with bilinear sampling. We applied this loss to the generated RGBD images conditioned on $c_1$ and $c_2$. The main difference between depth estimation (Zhou et al., 2017) and the proposed method is that our method optimizes both the RGB and depth image generator, while depth estimation only optimizes the depth estimator.

Moreover, for the constraints of the depth map in Equation 4, we define a consistency loss on the generated depth maps. This loss, which is similar to the left–right disparity consistency loss in (Godard et al., 2017), attempts to equate the depth map generated from $c_1$ to that generated from $c_2$ in 3D space. The overall proposed 3D loss function can be written as Equation 5.

$$\mathcal{L}_{3D} = \mathbb{E}_{z \sim p(z), c_{1,2} \sim p(c)} \left[ \frac{1}{3HW} ||G_{RGB}(z, c_1) - \text{warp}(G_{RGB}(z, c_2), c_{1 \to 2})||_1^1 \right.$$

$$\left. + \frac{1}{HW} ||\text{projection}(G_D(z, c_1), c_{1 \to 2}) - \text{warp}(G_D(z, c_2), c_{1 \to 2})||_1^1 \right] \quad (5)$$

Here, $G_{RGB}(z, c)$ and $G_D(z, c)$ denote the generated RGB and depth image from a latent vector $z$ and camera parameters $c$ respectively, $W$ and $H$ denote the width and height of the images respectively, and $c_{1\to 2}$ is a relative transformation matrix from $c_1$ to $c_2$. The "projection" operation calculates the depth value viewed from different viewpoints from the input depth map using Equation 3. For simplification, we omit the loss for the inverse transformation $c_{2\to 1}$ in the equation. The detailed explanations of "warp" and "projection" are provided in the appendix.

This loss function causes inaccurate gradients for the occluded pixels during the transformation $c_{1\to 2}$ because it does not consider those regions. Therefore, in this study, we used the technique proposed in (Gordon et al., 2019). This technique propagates gradients only to pixels where the projected depth is smaller than the depth of the other viewpoint image. This prevents inaccurate gradients in pixels that move behind other pixels during projection.

Finally, we add a depth constraint term to stabilize the training. The loss above can be easily minimized to 0 when the generated depth is extremely small. Therefore, we set the minimum limit for the depth value as $\mathcal{D}_{min}$ and add a regularization for depth values smaller than $\mathcal{D}_{min}$.

$$\mathcal{L}_{depth} = \frac{1}{HW} \sum_{x,y} \max\left(0, \mathcal{D}_{min} - \mathcal{D}(x, y)\right)^2 \qquad (6)$$

### 3.2.3 RGB image discriminator

To achieve the training of an RGBD generator from unlabeled RGB images, we apply adversarial loss only for the RGB channels of generated images. Although the loss can only improve the reality of the images, this loss is beneficial for learning depth images and camera parameter conditioning through the optimization of the loss in Equation 5.

Based on the above, the final objective for the generator $\mathcal{L}_G$ is as follows.

$$\mathcal{L}_G = \mathcal{L}_{GAN} + \lambda_{3D}\mathcal{L}_{3D} + \lambda_{depth}\mathcal{L}_{depth} \qquad (7)$$

Here, $\mathcal{L}_{GAN}$ is an adversarial loss function, and $\lambda_{3D}$ and $\lambda_{depth}$ are hyperparameters.

## 4 Experiments

### 4.1 Model architectures

The proposed method does not restrict the generator architecture: any type of image generators can be conditioned on camera parameters. To demonstrate the effectiveness of our method, we tested three types of image generation models: PGGAN, StyleGAN, and DeepVoxels. The model architectures are shown in Figure 3. Because perspective information is difficult to obtain from a single image, in this experiment, the camera intrinsics $K$ are fixed during training. We controlled only the azimuth $\theta_a$ (left–right rotation) and elevation $\theta_e$ (up–down rotation) parameters based on the training setting of HoloGAN. In the following, we provide the details of each model architecture.

**PGGAN:** PGGAN (Karras et al., 2018) is a state-of-the-art DCGAN. In this experiment, we conditioned the model on two camera parameters, azimuth and elevation, as follows: First, these values are input to cos and sin functions, respectively, and the outputs are concatenated to a single four–dimensional vector $c_{cyclic}$. Subsequently, $c_{cyclic}$ is concatenated to the latent vector $z$, which is input to the generator. This operation allows the generated images to change continuously for a $360°$ angle change. We start with a resolution of $32 \times 32$ and increase it progressively to $128 \times 128$.

**StyleGAN:** StyleGAN (Karras et al., 2019) is a state-of-the-art GAN model that controls the output "style" of each convolutional layer by performing adaptive instance normalization (AdaIN) (Huang & Belongie, 2017) and acquires hierarchical latent representations. We used $c_{cyclic}$ to only control the style of features on resolutions of $4 \times 4$ and $8 \times 8$, as it is known that styles at low-resolution layers control global features such as the pose and shape of an object. More concretely, we concatenated $c_{cyclic}$ and the output of the mapping network $w$, which was then converted to $w'$ with a multilayer perceptron. Please refer to Figure 3. The image resolution is the same as PGGAN.

**DeepVoxels:** HoloGAN enables the disentanglement of camera parameters by using 3D latent feature representations. This is more natural modeling of the 3D world than the two models above because it considers explicit transformations in 3D space. However, HoloGAN cannot consider depth

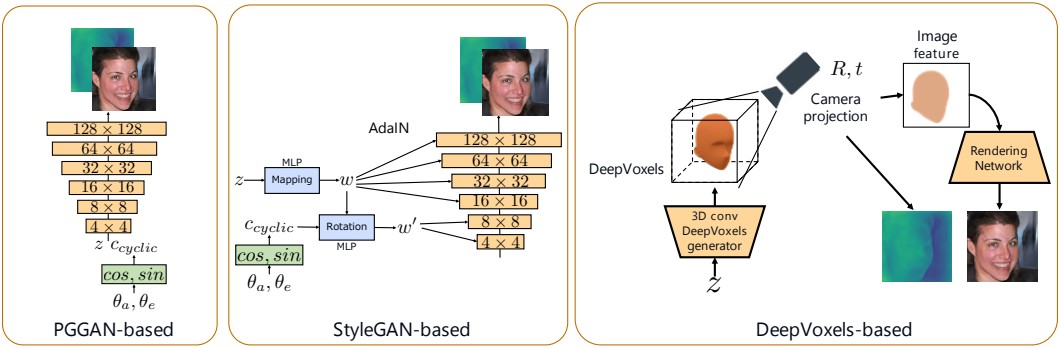

Figure 3: Generator architectures tested. PGGAN-based model (left), StyleGAN-based model (middle), and DeepVoxels-based model (right).

information as the projection unit of HoloGAN only calculates the weighted sum of the feature on the depth dimension. Therefore, we used the model inspired by DeepVoxels (Sitzmann et al., 2019) to apply the proposed method. DeepVoxels is a method that can learn the 3D latent voxel representation of objects using images from multiple viewpoints of a single object; additionally, it can generate novel-view images. This method uses the occlusion-aware projection module that learns which voxels are visible from the camera viewpoint along the depth axis. This is achieved via unsupervised learning. Therefore, a depth image can be acquired from the model, which is suitable for combining with our method. In this experiment, we combined DeepVoxels and a voxel feature generator that generates features from random latent vector $z$, for the random image generation task. We used 3D convolution and AdaIN for the voxel feature generator, similarly to HoloGAN. DeepVoxels uses an explicit camera model to acquire the feature visible in the camera frustum, whereas HoloGAN uses rigid-body transformations. Therefore, DeepVoxels enables more accurate reasoning about the 3D world. We compare the three settings for the models using 3D feature representations. The first model uses the weighted sum on the depth dimension instead of occlusion-aware projection modules, similarly to HoloGAN. The second model uses occlusion-aware projection modules but does not use the proposed 3D loss. The final model uses DeepVoxels and the proposed 3D loss. The methods are called "HoloGAN-like," "DeepVoxels," and "DeepVoxels + 3D loss" in the figures and tables. It is noteworthy that "HoloGAN-like" is not the same model as the original HoloGAN because it is based on DeepVoxels' network structures.

## 4.2 DATASETS

We trained our model using FFHQ (Karras et al., 2019), cars from ShapeNet (Chang et al., 2015), car images (Krause et al., 2013), and the LSUN bedroom dataset (Yu et al., 2015). We used 128 × 128 images for the PGGAN and StyleGAN, and 64 × 64 images for models using 3D latent feature representations owing to memory constraints. We used 35° for the elevation angle range for all experiments, 120° for the azimuth range for the FFHQ and bedroom datasets, and 360° for the azimuth range for the Car and ShapeNet car datasets. For the ShapeNet car dataset and car image dataset, we used a new occlusion reasoning algorithm for DeepVoxels–based models to stabilize the training. The details are explained in the appendix.

## 4.3 RESULTS

**Qualitative results** The generative results from each model controlling the camera parameters on the FFHQ and ShapeNet car datasets are shown in Figures 4, 5, 10, and 11. In the figures, images with colormaps show the generated depth images. The depth is normalized (subtracted by the minimum value and divided by the range) and visualized with colormaps. For all models using the proposed loss (top three in the figures), images can be generated by controlling the camera parameters while preserving their identity. Moreover, the models can generate depth images that do not exist in the training samples. To confirm the depth consistency, we show normal maps and rotated images for the generative results from each model, as shown in Figure 6. The white regions of the

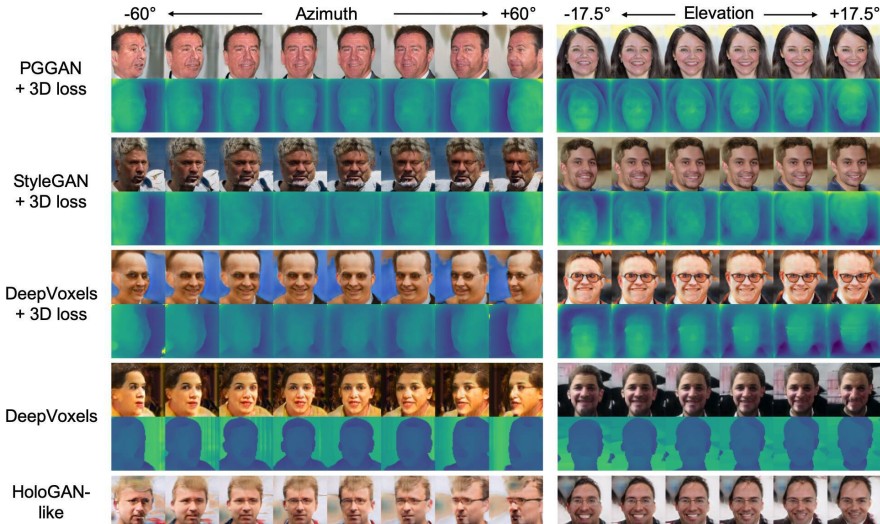

Figure 4: Visualization of comparison for the generated images from each model on FFHQ dataset. Images in each row are generated from the same latent vector $z$ but different azimuth or elevation angles. The images with colormaps are the generated depth images.

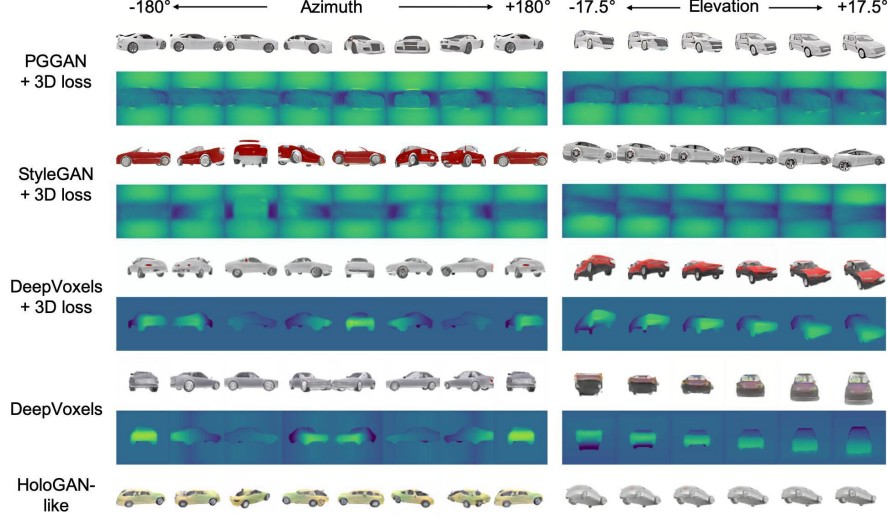

Figure 5: Visualization of comparison for the generated images from each model on ShapeNet car images. Images in each row are generated from the same latent vector $z$ but different azimuth or elevation angles. The images with colormaps are the generated depth images.

ShapeNet car dataset are omitted for the visualization of point clouds. As shown in the figure, the models can generate the convex shape of a face and the rectangular shape of a car without any annotations regarding the 3D world. In particular, although the PGGAN and StyleGAN use a 2D CNN, consistent rotation and depth estimation are achieved, which is impossible with previous methods. This implies that the proposed method has good generalization performance on the generator architecture. The DeepVoxels–based method with the proposed loss performs well on both FFHQ and the ShapeNet car dataset. They can acquire more consistent rotation and generate more consistent depth images than 2D CNN–based models. This is thanks to the explicit 3D space modeling, though it does consume much memory and has high computational cost. In the experiments, although the output images for DeepVoxels–based models were half the size than that of StyleGAN–based models, they required 2.5 times higher memory and 2.9 times longer computational times for one iteration.

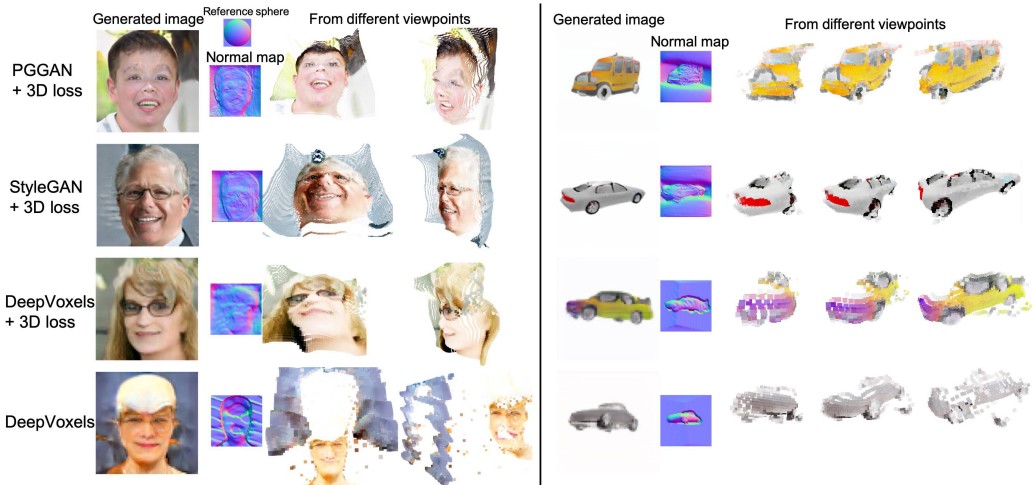

Figure 6: Normal map and point cloud visualization for FFHQ and ShapeNet car datasets. Point clouds in occluded region are not visualized in the figure.

Table 1: Performance comparison of unconditional generation models and proposed camera parameter–conditional models. We report FID, $V_{depth}$, and $V_{color}$ (lower is better) for each model.

| METRICS | FFHQ | | | ShapeNet Car | | |
|---|---|---|---|---|---|---|
| | FID | $V_{depth}$ | $V_{color}$ | FID | $V_{depth}$ | $V_{color}$ |
| PGGAN | **28.5** | - | - | 16.7 | - | - |
| PGGAN + 3D loss | 30.3 | 0.00141 | 0.0142 | **14.5** | 0.00043 | 0.0444 |
| StyleGAN | **20.9** | - | - | 15.5 | - | - |
| StyleGAN + 3D loss | 24.2 | 0.00077 | 0.0092 | **13.5** | **0.00027** | 0.0469 |
| HoloGAN-like | 23.4 | - | - | 33.5 | - | - |
| DeepVoxels | **19.4** | 0.00487 | 0.0153 | **28.6** | 0.00045 | 0.0398 |
| DeepVoxels + 3D loss | 21.1 | **0.00067** | **0.0072** | 31.2 | **0.00030** | **0.0283** |

For the ShapeNet car dataset in Figure 5, PGGAN- and StyleGAN-based methods can generate consistently rotated images. However, for the PGGAN, only a $180°$ azimuth change is acquired. This is because the model cannot distinguish between the front and back of the car, as it is difficult to achieve only with unsupervised learning. Meanwhile, StyleGAN-based methods can learn consistent azimuth and elevation angle changes. This is because the StyleGAN is stable owing to its hierarchical latent representation.

Here we will compare the three 3D–latent–feature–based methods. In our training settings, the "HoloGAN–like" method works well on the FFHQ dataset but cannot acquire consistent $360°$ rotation on the ShapeNet car dataset. DeepVoxels–based methods, on the other hand, can control $360°$ object rotation on the dataset, realizing the depth map generation without any supervised information. This result shows that the depth reasoning helps to generate images considering the 3D geometry of the objects. Moreover, DeepVoxels–based method with the proposed loss can generate more consistent images for the FFHQ dataset. For example, in "DeepVoxels", the depth of the background is smaller than that of the face, and the background pixels hide foregrounds when the generated images are rotated as shown in Figure 6. However, this is not observed with the proposed loss, as our method considers warped images from different viewpoints, which facilitates learning the 3D world accurately.

Moreover, additional generative results on the car image and bedroom datasets are depicted in Figure 7. These datasets are more difficult to train than the other two datasets due to the imbalanced distribution of the camera poses and the diversity of the object layouts. Although the models can learn consistent rotation reasonably for both datasets, the models cannot generate consistent depth maps. The results show the difficulty of the unorganized datasets.

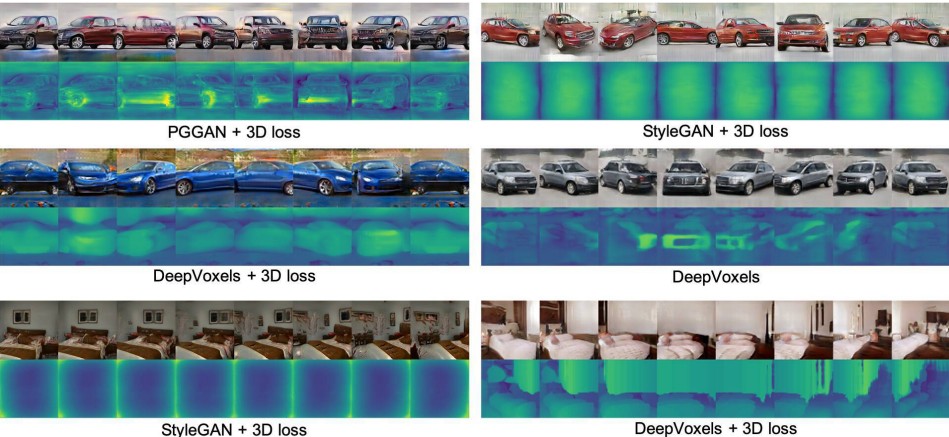

PGGAN + 3D loss                StyleGAN + 3D loss

DeepVoxels + 3D loss                DeepVoxels

StyleGAN + 3D loss                DeepVoxels + 3D loss

Figure 7: Generated car and bedroom images changing the azimuth angle range.

As a result, the proposed method effectively helps various generators to learn both depth information and explicit controls on camera poses. These are achieved without the need for 3D latent representations as required in HoloGAN. Moreover, the proposed method further improves the results for the models using 3D latent representations.

**Quantitative evaluation on RGB images**   We compared the Fréchet inception distance (FID) (Heusel et al., 2017) between models with and without the proposed method for each generator architecture. FID is a typical evaluation metric for the quality and diversity of the generated RGB images. The results are shown in Table 1. The results show that the proposed camera parameter–conditional image generation models can generate images with comparable or even better FIDs than unconditional or RGB image generation models for all generator architecture types. Notably, this is an unfair comparison because the models without 3D loss were trained just to minimize the distribution distance between the appearance of the real and generated images, although the models with 3D loss learn camera parameter conditioning. The results show the robustness and effectiveness of our method against the generator architectures.

**Quantitative evaluation on RGBD images**   Evaluating the generated RGB and depth in the 3D space is difficult as obtaining the ground truth color or depth for the generated images is impossible. A possible approach to evaluate RGBD images without ground truth images is by calculating the inception score (IS) (Salimans et al., 2016) or FID on the generated images. However, this is inappropriate, as IS and FID are estimated in the feature space of a pre-trained CNN, and they cannot consider the 3D world geometry. Therefore, the evaluation of the generated RGBD with the 3D space is unattainable. Instead, we evaluated the color and depth consistency across the views to quantitatively compare the RGBD images generated by different methods. For the point clouds generated from the same latent vector $z$, but different camera parameters $c$, all points should be on a single surface in the 3D space. Therefore, by calculating the variation of the generated RGBD across the views, the 3D consistency can be quantitatively evaluated. We calculated the variance of the point clouds for generated images as $V_{depth}$ and $V_{color}$ in Table 1. The details are provided in the appendix. notably, these values cannot be calculated for PGGAN and StyleGAN without 3D loss, and HoloGAN–like method.

The results show that DeepVoxels with 3D loss exhibited the best scores, due to the strong inductive bias of the 3D–latent–representations. Compared to DeepVoxels without 3D loss, DeepVoxels with 3D loss generated consistent color and depth, exhibiting the effectiveness of the proposed loss.

## 5   CONCLUSION

We herein proposed an RGBD image synthesis technique for camera parameter–conditional image generation. Although the proposed method does not require any labeled dataset, it can explicitly con-

trol the camera parameters of generated images and generate consistent depth images. The method does not limit the generator architecture, and can be used to condition any type of image generator on camera parameters. As the proposed method can learn the relationship between camera parameters and images, future works will include extending the method for unsupervised camera pose estimation and unsupervised camera pose invariant feature extraction from images.

## 6 ACKNOWLEDGEMENT

This work was partially supported by JST CREST Grant Number JPMJCR1403, and partially supported by JSPS KAKENHI Grant Number JP19H01115. We would like to thank Antonio Tejero de Pablos, Dexuan Zhang, Hiroaki Yamane, James Borg, Mikihiro Tanaka, Sho Maeoki, Shunya Wakasugi, Takayuki Hara, Takuhiro Kaneko, and Toshihiko Matsuura for helpful discussions.

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

## A  "WARP" AND "PROJECTION"

Here, we explain the "warp" and "projection" operation in Equation 5.

**Warp**  When we warp $G_{RGB}(z, c_2)$ to $G_{RGB}(z, c_1)$, first we calculate the position of pixels $p_{c_1}$ in $G_{RGB}(z, c_1)$ when they are viewed from $c_2$.

$$\mathcal{D}_{c_2} p_{c_2} = K R_{1 \to 2} K^{-1} \mathcal{D}_{c_1} p_{c_1} + K t_{1 \to 2} \tag{8}$$

Here, $R_{1 \to 2}$ and $t_{1 \to 2}$ are relative rotation and translation matrices respectively. We warp $G_{RGB}(z, c_2)$ according to the calculated positions, $p_{c_2}$. The operation is implemented with bilinear sampling between the four neighboring pixel colors of warped coordinates, such that the operation is differentiable. Same operation is performed to warp generated depth images.

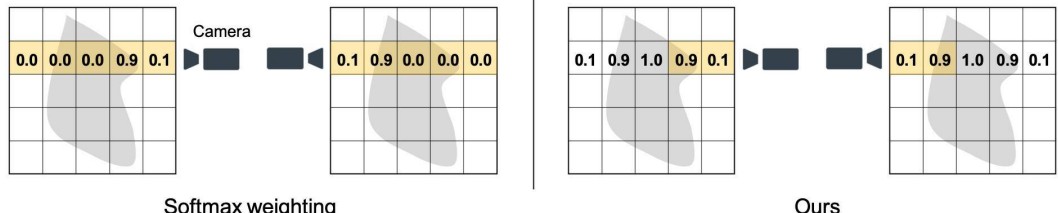

Figure 8: Comparison between the softmax weighting (left) and our occlusion reasoning algorithm (right). The values denote the voxel weight, and the orange regions are the visible voxels in the camera. For softmax weighting, the occlusion network needs to change the voxel weight, according to the camera location. Our method accumulates the weight along the camera ray and ignored the voxels where the accumulative values exceed one, thus the occlusion network does not need to change the weights according to the camera location.

**Projection**  As the depth values in $\text{warp}(G_D(z, c_2), c_{1 \to 2})$ are sampled from $G_D(z, c_2)$, which is viewed from $c_2$, to compare $G_D(z, c_1)$ with $\text{warp}(G_D(z, c_2), c_{1 \to 2})$, we need to project the depth values of each pixel in $G_D(z, c_1)$ to the viewpoint $c_2$. This operation is the same as in Equation 8, and $\mathcal{D}_{c_2}$ is used for the projected depth values. We denote it as "projection".

## B  $V_{depth}$ AND $V_{color}$

These metrics evaluated the depth and color consistency across the views. First, the images were generated from the same $z$ but different $c$. Second, the point clouds were plotted for each image in the real-world coordinate. Third, the coordinates were converted to polar coordinates from the origin, and the angle coordinates (azimuth and elevation) were quantized. Fourth, for each image, the points were aggregated following the quantized coordinates, which we denote as "cell". For each cell, the point with the smallest radial coordinates were sampled. This prevented inaccurate variance for cells with multiple surfaces. Fifth, the variance of the depth and color for each cell, across different $c$, was calculated. Sixth, the variance was averaged across cells and different $z$. Randomly sampled 100 $z$ and 100 $c$ for each $z$ were employed to calculate the metrics.

For face images, the origin was set as $(0, 0, -0.5)$ (behind the head), and used $-23°$ to $23°$ for angular coordination. For ShapeNet car dataset, the origin was set as $(0, 0, 0)$, and used $-180°$ to $180°$ for the azimuth angle and $-90°$ to $90°$ for the elevation angle. The white region was ignored for the evaluation of the ShapeNet car images.

## C  CAMERA PARAMETER DISTRIBUTIONS

To randomly sample two similar camera parameters, $c_1$ and $c_2$, we sampled $c_1$ from a uniform distribution and sampled $c_2$ from an area near to $c_1$, within the angle range. We limited the maximum distance between $c_1$ and $c_2$ as $30°$, to avoid large occlusions. Since it is difficult to obtain camera intrinsics $K$ from a single image, we fix $K$ and the camera distance during training. We used $K$ as,

$$K = \begin{pmatrix} 2s & 0 & \frac{s}{2} \\ 0 & 2s & \frac{s}{2} \\ 0 & 0 & 1 \end{pmatrix}, \tag{9}$$

wherein $s$ is the size of images. We fixed the distance between the camera and origin of coordinates as 1.

## D  IMPLEMENTATION DETAILS FOR DEEPVOXELS

The DeepVoxels–based methods were implemented using the projection layer, occlusion module, and rendering module proposed in (Sitzmann et al., 2019). We implemented them with simpler

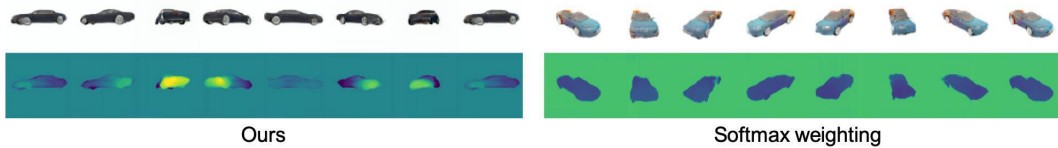

Figure 9: Comparison of the occlusion reasoning algorithms. The proposed method can acquire more consistent rotation and generate consistent depth maps than the softmax weighting.

structures than those in the original implementation, to reduce the computational and memory expense. We used fewer 3D convolutional layers for the occlusion module and a U-Net-like network with AdaIN for the rendering module. Moreover, for simplicity, we did not use the Identity regulariser or style discriminator proposed in (Nguyen-Phuoc et al., 2019).

We employed a different algorithm in occlusion reasoning to enable consistent depth for the ShapeNet car and car image datasets. The occlusion reasoning used to get image features in Deep-Voxels is softmax weighting along the depth axis, which is visualized on the left side in Figure 8. This algorithm needs the occlusion module to calculate the weight of the voxels according to the camera poses, which is difficult through unsupervised learning. Therefore, to reduce the training expenses of the occlusion network, we employ an explicit reasoning algorithm. The network estimates the probability of each voxel to be on the surface of the object, i.e., the opacity of each voxel. This is implemented using a sigmoid activation function. Further, the weights are accumulated along the rays from the camera by adding-up the values. When the accumulated values exceed 1, the later voxels are ignored by replacing the weight values with 0s. By doing this, the occlusion module does not need to change voxel weight according to the camera poses. The algorithm overview is shown in the right side of Figure 8.

The image generation results on the ShapeNet car dataset using "DeepVoxels + 3D loss", with each algorithm, are depicted in Figure 9. The proposed occlusion reasoning model can acquire more consistent $360°$ rotation, whereas the softmax weighting cannot. Moreover, the proposed algorithm can generate consistent depth maps compared to the softmax weighting method. The results show the effectiveness of the proposed method for unsupervised learning.

# E    TRAINING DETAILS

We trained PGGAN– and StyleGAN–based models for 250,000 iterations using batch-size of 32, and 3D–latent–feature–based models for 65,000 iterations with a batch-size of 10. All models are trained with Adam optimizer with equalized learning rates 0.001 for the generators, 0.00001 for the mapping networks, and 0.003 for the discriminators. In the experiments, we used a ResNet-based discriminator and non-saturating loss (Goodfellow et al., 2014) with gradient penalty (Mescheder et al., 2018). Training with a single NVIDIA P100 GPU required 30, 40, and 50 hours for DeepVoxels–, StyleGAN–, and PGGAN–based methods, respectively.

# F    ADDITIONAL RESULTS

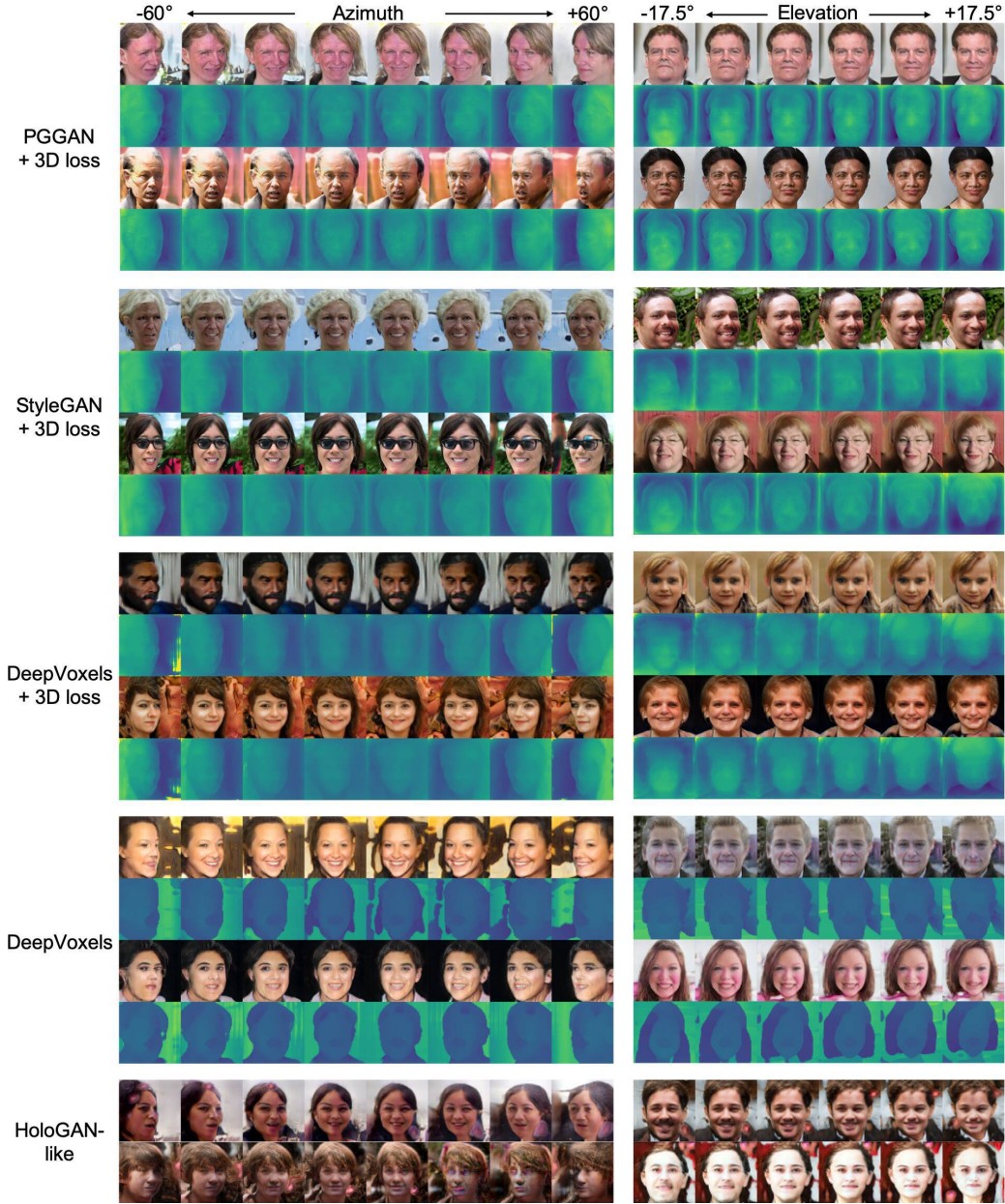

Figure 10: Additional results on FFHQ dataset. Images in each row are generated from the same latent vector $z$ but different azimuth or elevation angles. The images with colormaps are the generated depth images.

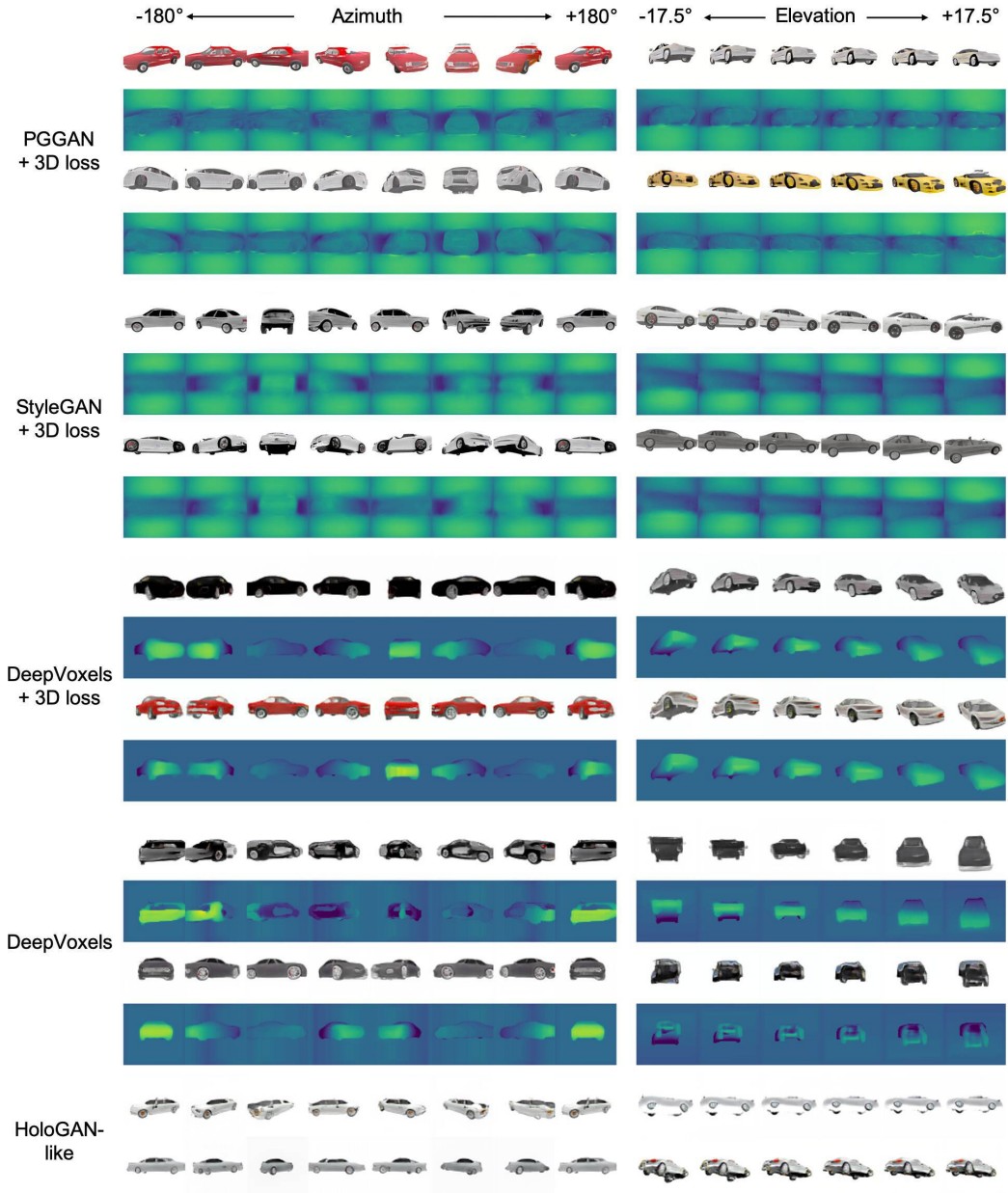

Figure 11: Additional results on ShapeNet car dataset. Images in each row are generated from the same latent vector $z$ but different azimuth or elevation angles. The images with colormaps are the generated depth images.

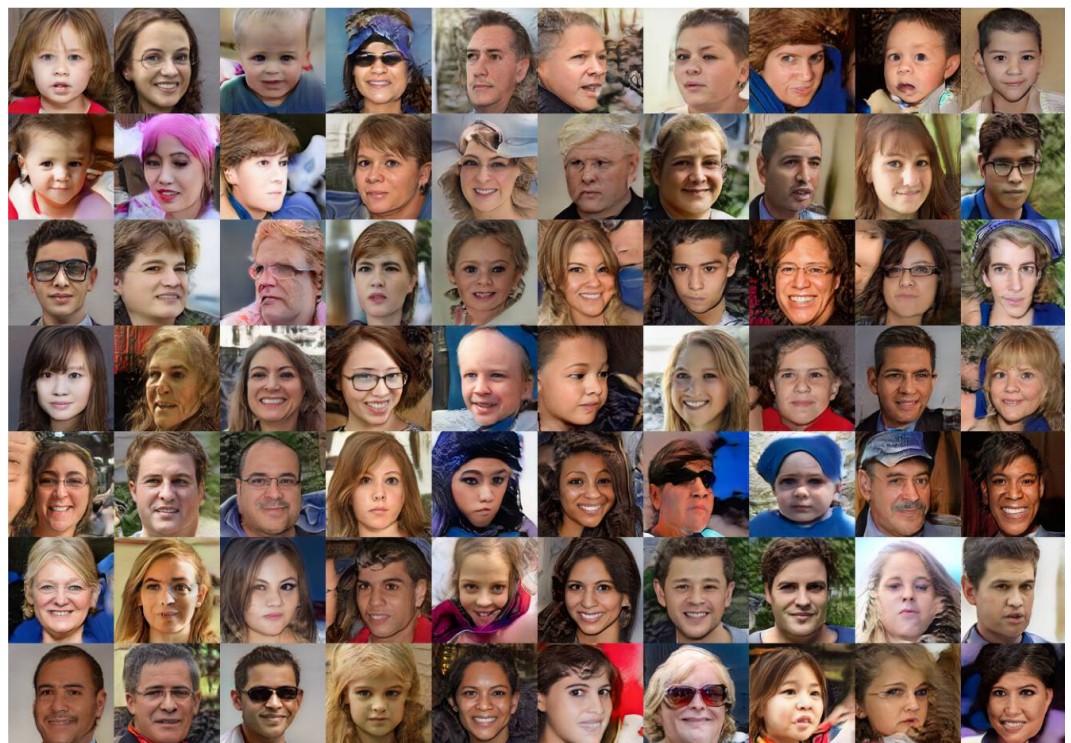

PGGAN + 3D loss

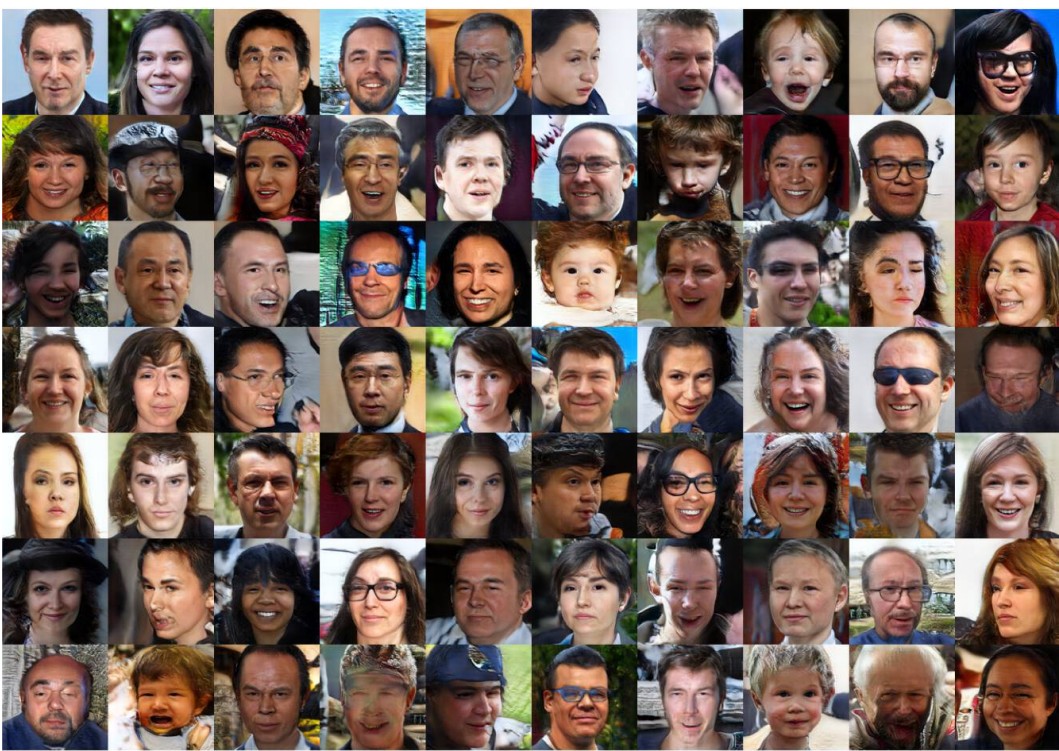

PGGAN

Figure 12: Randomly generated RGB images on the FFHQ dataset from PGGAN, with and without proposed loss.

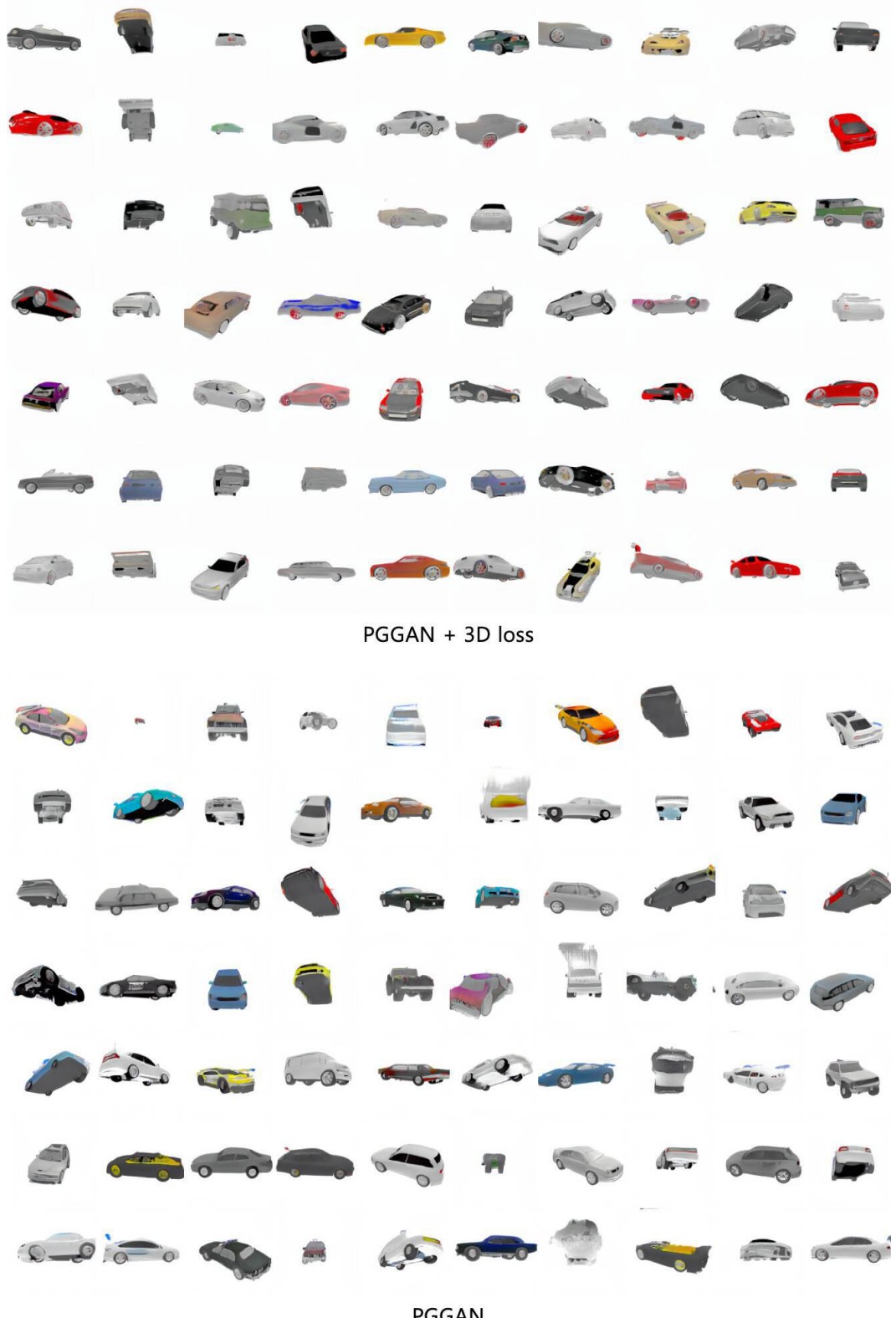

PGGAN + 3D loss

PGGAN

Figure 13: Randomly generated RGB images on the ShapeNet car dataset from PGGAN, with and without proposed loss.

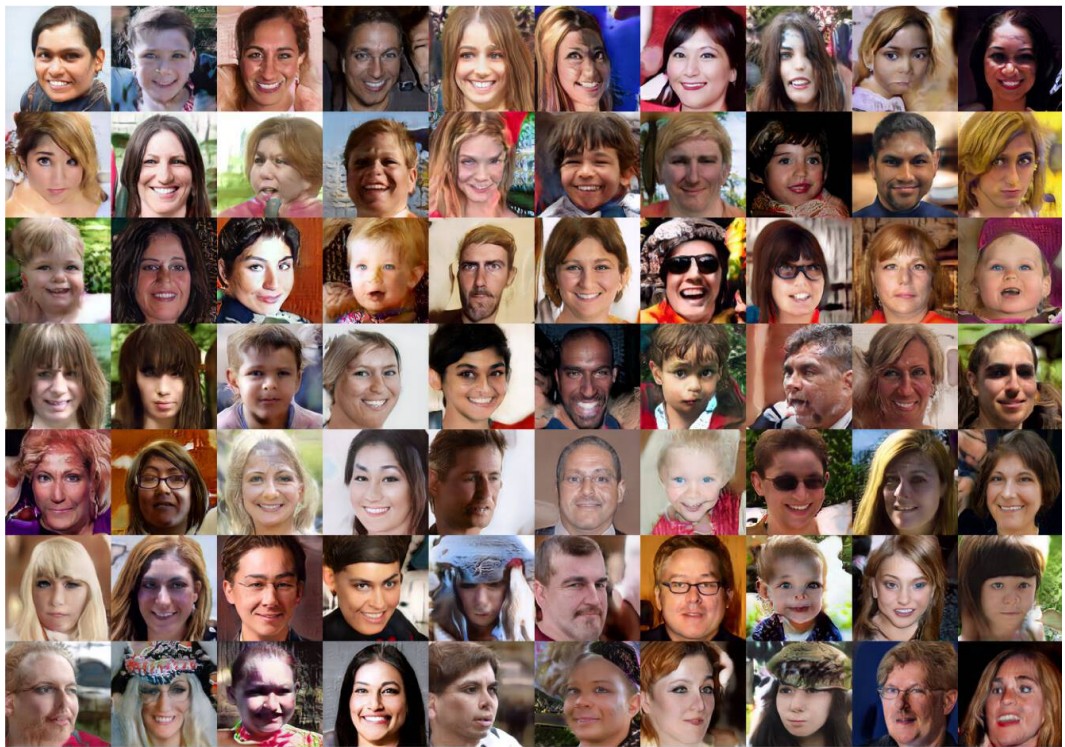

StyleGAN + 3D loss

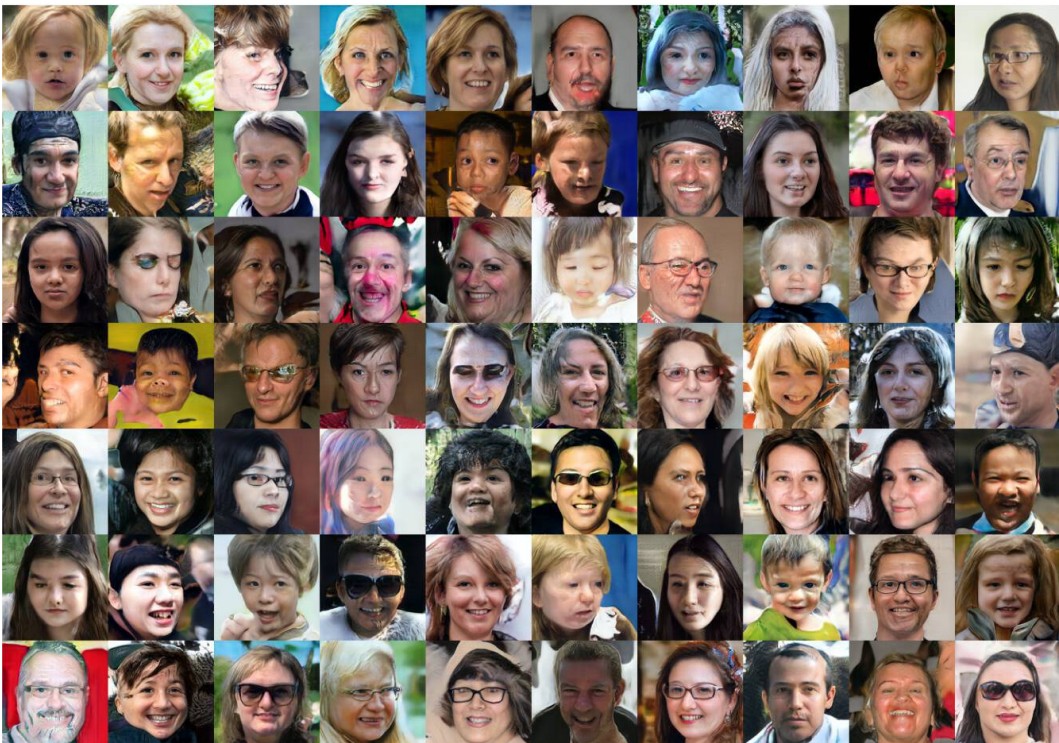

StyleGAN

Figure 14: Randomly generated RGB images on the FFHQ dataset from StyleGAN, with and without proposed loss.

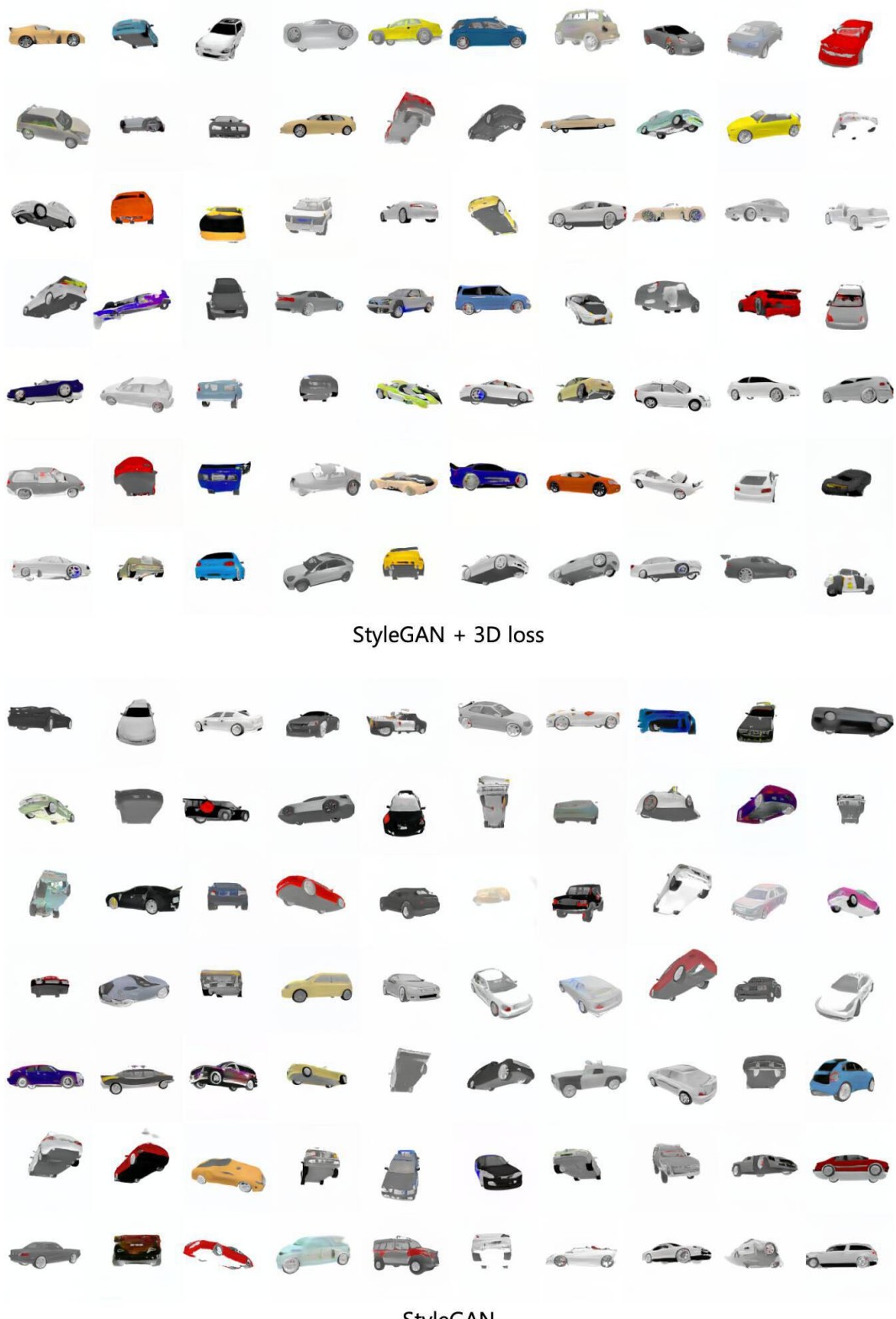

Figure 15: Randomly generated RGB images on the ShapeNet car dataset from StyleGAN, with and without proposed loss.

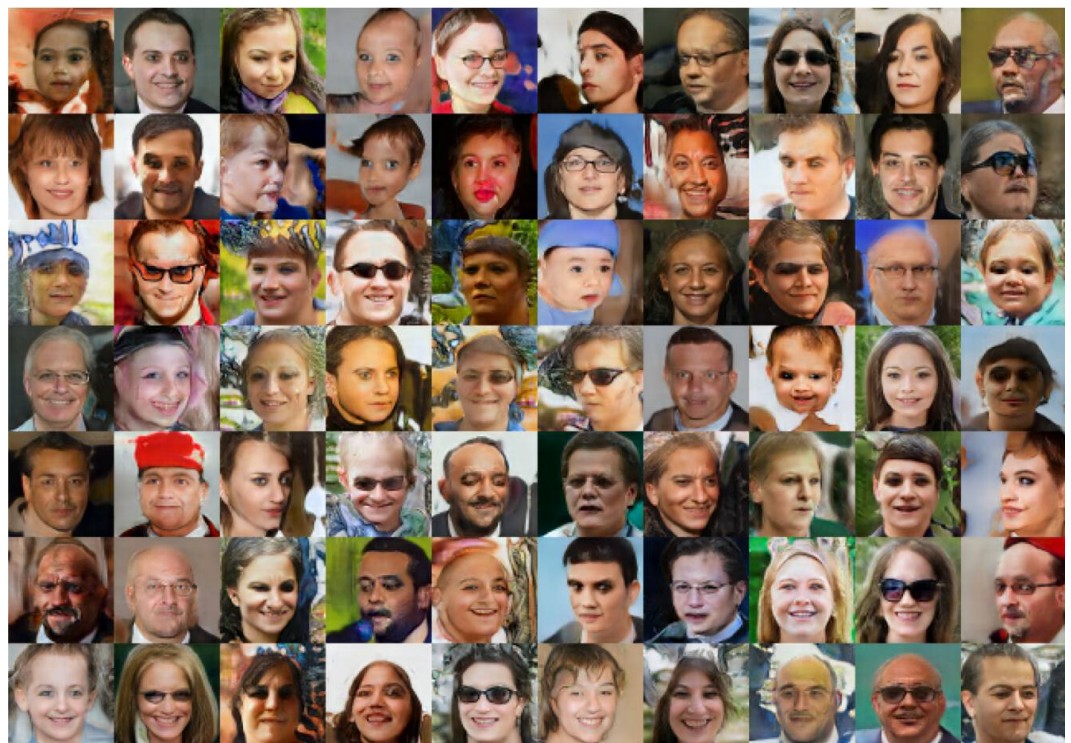

DeepVoxels + 3D loss

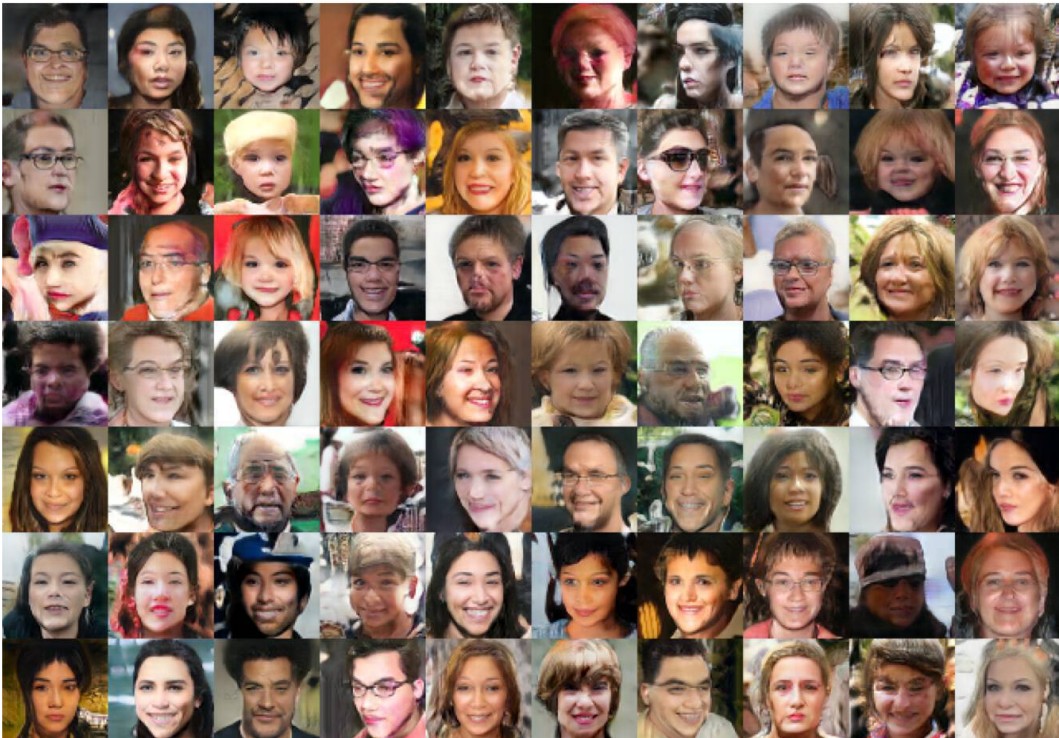

DeepVoxels

Figure 16: Randomly generated RGB images on the FFHQ dataset from DeepVoxels, with and without proposed loss.

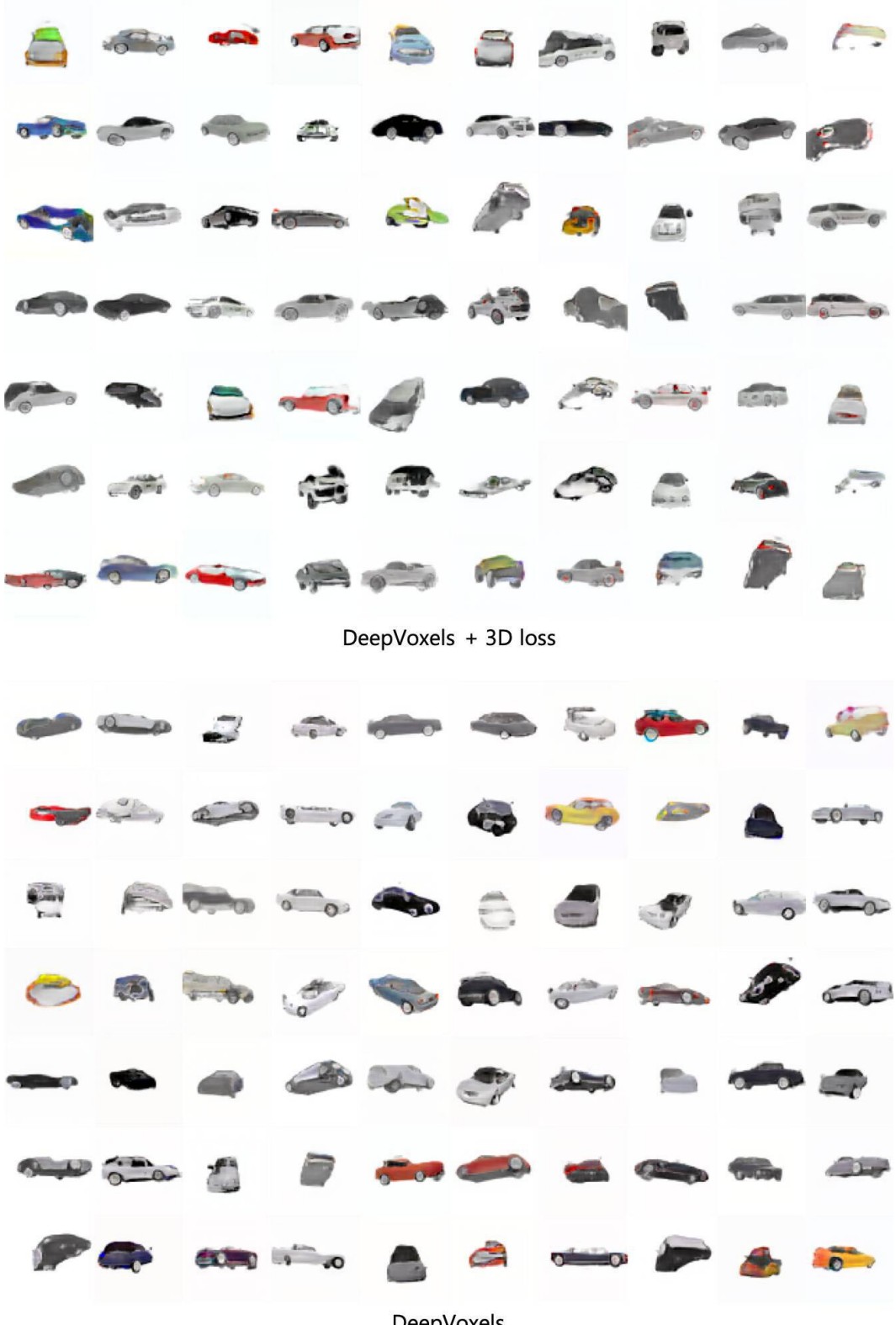

Figure 17: Randomly generated RGB images on the ShapeNet car dataset from DeepVoxels, with and without proposed loss.

