# OpenReview forum: "RGBD-GAN: Unsupervised 3D Representation Learning From Natural Image Datasets via RGBD Image Synthesis"
_ICLR.cc/2020/Conference — Accept (Poster)_

### Official Review · AnonReviewer1 · 2019-10-21
**Official Blind Review #1**

**Rating:** 6

**Review:**

# Review ICLR20, RGBD-GAN

This review is for the originally uploaded version of this article. Comments from other reviewers and revisions have deliberately not been taken into account. After publishing this review, this reviewer will participate in the forum discussion and help the authors improve the paper.

## Overall

The article proposes a method of modifying image-generating networks to also produce depth maps in an unsupervised way by enforcing rotational consistency.

I enjoyed reading this work and I'm recommending it to be accepted. However, first there are some (in my opinion straight-forward) changes that need to be made to this work before I can recommend its publication:

- The common "Related Works" section is missing and some of the literature is taking place in the introduction. I find this unorganized and I'd recommend keeping the intro shorter and just moving the literature either behind the intro or to the end of the paper.
- Most figures and especially your headline figure (1) suffer from not having the depth normalized and not having a scale to it. The fix for this is simple and two-fold: for each depth image, subtract the minimum value and divide by the range (to normalize it and increase contrast), then write in the caption or as a legend that white is closer to the camera and black is further back.
- 3D vs. 2.5D - If the common geometric definition of "3D" was applied here, the article's title was correct. However, in computer vision and especially 3D vision, the term is commonly used to refer only to models that include full scene geometry, including the occluded backs of objects and the term 2.5D is used to describe assigning depth values to pixels in an RGB image (and therefore only covering the view-dependent front of the object), which I think is the case here. However, this is not a hill that I'll die on so if you insist on that terminology, I won't block acceptance.
- When you first discuss HoloGAN, you mention one of its main downsides being scalability and then proceed to not only explain that but also use a HoloGAN-like architecture in one of your experiments. I'd either remove the scalability argument or justify not just that but also how that's not relevant to your experiments.
- The following phrase occurs multiple times throughout: "camera parameter conditional image generation". I _think_ you're missing a dash between "parameter" and "conditional".


## Specific comments and questions

### Abstract

All good.

### Intro

- Fig.1 normalize image
- The literature section in intro mentions "For all methods, 3D annotations must be used..." - that's not true. See [Rezende, 2016][1] and [Rajeswar, 2018][2]
- I understand how some literature is required to position your method, but I think it's better to not have the entire literature section in the center of the introduction

[1]: https://arxiv.org/abs/1607.00662
[2]: https://openreview.net/forum?id=BJeem3C9F7

### Method

- 2.1 clear + nicely written
- Figure 2 good, caption a bit too short - figure+caption should be able to stand on their own
- Illustration of Figure 3 nice, except for unclear DeepVoxel part: what's the wavy orange flag stand for?

### Experiments

- You mention K is fixed, but where does the initial K come from? I assume it's just neglected (since it's not important for StyleGAN/PGGAN), but then this needs to be mentioned in the methods sections closer to the formulas dealing with K.
- Figure 4 - the depth maps need to be normalized. All we see here is a grey mush, even worse in Fig. 7
- For ShapeNet cars, the model seems to suffer from not having a reference for the top and bottom of the image - have you tried adding floor/sky?
- Figure 6, the tire marker is a good idea but image still unclear - I recommend slightly less rotation or an intermediate step between generated image and e.g. front view
- For quantitative results/FID: try using Hausdorff or Chamfer distance on the rendered scenes' pixels. We don't care about the goodness of the RGB generation but the depth.

### Conclusion

All good, albeit a bit short.

### Appendix

I don't think I saw any references to the appendix in the main paper.

**Experience Assessment:**

I have published one or two papers in this area.

**Review Assessment: Checking Correctness Of Derivations And Theory:**

I assessed the sensibility of the derivations and theory.

**Review Assessment: Checking Correctness Of Experiments:**

I carefully checked the experiments.

**Review Assessment: Thoroughness In Paper Reading:**

I read the paper thoroughly.

---

> ### Author Response · Authors · 2019-11-13
> **Response to Reviewer #1**
>
> We would like to thank the reviewer for valuable comments.
>
> Related works
> - We will separate the related work section from the introduction section.
>
> Depth visualization
> - We will normalize the depth maps and visualize it in colormap (as reviewer2 says) for better visualization.
>
> 3D vs. 2.5D
> - Our model can not only generate RGBD images, which is commonly considered 2.5D, but also explicitly control camera poses while preserving the image content. Therefore, it can be regarded that the model can learn full 3D geometry implicitly, though the output is not fully 3D. This is the intuition to use the word "3D".
>
> Scalability of HoloGAN
> - We agree that the badness of the scalability of HoloGAN is not supported by our experimental results. Therefore, we will delete the scalability part from the introduction.
>
> Necessity for 3D annotations
> - Thank you for introducing related papers. Though they do not need annotations, both methods can only deal with synthetic primitive datasets. Our method, however, can work on natural images. We will add the discussion to the related works section.
>
> wavy flag
> - This is a conceptual figure of learned DeepVoxels. DeepVoxels are implicit representations, and we cannot visualize what is acquired. We agree that the figure is ambiguous, we will replace the figure.
>
> About K
> - Because learning K from single images is difficult, we initialize K with [[2s, 0, s/2], [0, 2s, s/2], [0, 0, 1]] (numpy-style order), where $s$ is image size. We will add the explanation in the paper.
>
> floor/sky
> - We did not try adding floor or sky to render the ShapeNet car dataset. We think adding simple sky or floor will help learning depth information to some extent, but it is difficult to learn consistent depth. This is because foreground regions have common salient concepts across views (eg. tire, headlight, window, ...) but the background does not. This is also problematic when we train the model on a car image dataset, which has floor and sky, as shown in Figure 7.
>
> Evaluation for depth
> - Evaluating the generated depth is difficult because we cannot obtain ground truth depth for the generated images. A possible approach to evaluate depth images without ground truth images is calculating the inception score (IS) [5] or FID on the generated depth images, but we do not think it is appropriate. This is because IS and FID are estimated in the feature space of a pre-trained CNN, and they cannot consider the geometry in the 3D world. Therefore it is almost impossible to evaluate how the generated depth is plausible in 3D space. Instead, we will evaluate the depth consistency across views to quantitatively compare the generated depth among different methods. When we plot point clouds generated from the same $z$ but different $c$, all points should be on a single surface. Therefore, by calculating the variation of the generated depth, we can quantitatively evaluate the 3D consistency across views. It is expected that 3D-latent-feature-based models have better performance than other models. We will add the results in the paper.
>
> [5] Tim Salimans, Ian Goodfellow, Wojciech Zaremba, Vicki Cheung, Alec Radford, and Xi Chen. Improved techniques for training gans. In NIPS, 2016.

---

### Official Review · AnonReviewer2 · 2019-10-23
**Official Blind Review #2**

**Rating:** 3

**Review:**

The submission proposes a technique to learn RGBD image synthesis from RGB images. A distinctive feature proposed by the technique is the user-controllable camera rotation parameters, learned in an unsupervised manner. The technique can be used in conjunction with various models, such as PGGAN, StyleGAN, and DeepVoxels.

This paper provides an interesting approach that can be a useful building block for future investigations.

The main issue I see with the paper is the number of results provided. Only 2 different images are shown per combination of model and dataset, limiting the reader's ability to assess the technique's performance. Would it be possible to provide a large number of results in a supplementary material or appendix?

In my opinion, this may be due to a difference in writing style, but the paper, in general, is slightly hard to read.

The depth in figures 1, 4, 5, 7 and 9 would be easier to read if it was displayed as a colormap (with the corresponding color bar) instead of grayscale. Additionally, a reference sphere would be appreciated near the normal maps shown in fig. 6 to inform the reader of the coordinates system used.

Sec. 3.3 states that “the depth of the background is smaller than that of the face [for DeepVoxels]; however, this does not occur when the proposed loss is used”, however fig. 6 seems to show the contrary. Is it due to the depth discontinuity?

Minor details
- Sec. 3 “Experimetns”: typo.
- Sec. 3.3, “[...] use the 2D CNN”: I would replace “the” by “a”.
- Sec. 3.3, Third paragraph, the first sentence is hard to read.
- Sec. B “the later voxels are ignore[d]”


**Experience Assessment:**

I have read many papers in this area.

**Review Assessment: Checking Correctness Of Derivations And Theory:**

I assessed the sensibility of the derivations and theory.

**Review Assessment: Checking Correctness Of Experiments:**

I assessed the sensibility of the experiments.

**Review Assessment: Thoroughness In Paper Reading:**

I read the paper at least twice and used my best judgement in assessing the paper.

---

> ### Author Response · Authors · 2019-11-13
> **Response to Reviewer #2**
>
> We would like to thank the reviewer for valuable comments.
>
> Additional results
> - We will add a large number of results to the appendix
>
> Writing style
> - We will get English proofreading for our paper. If you do not mind, please let us know concretely which parts are difficult to read.
>
> Depth visualization
> - For all depth images, we will normalize them (as reviewer1 says) and visualize them with colormap. Moreover, we will add a reference sphere in Figure 6.
>
> Background depth
> - Figure 6 shows the results on StyleGAN. The background depth seems small in Figure 6, but it does not cover foregrounds even if the image is rotated (within the angle range during training). This is contrary to the results of DeepVoxels in Figure 4, where the background pixels hide foregrounds when the generated image is rotated. I will add this explanation in Section 3.3, and comparison experiments in the appendix.

---

### Official Review · AnonReviewer3 · 2019-10-23
**Official Blind Review #3**

**Rating:** 6

**Review:**

SUMMARY: Unsupervised/Self-supervised generative model for image synthesis using 3D depth and RGB consistency across camera views

CLAIMS:
- New technique for RGBD synthesis using loss in 3D space
- Can disentangle camera parameters from content (I disagree slightly with "disentangle" since you are conditioning on camera parameters in the first place)
- Different generator architectures can be used

METHOD:
Generate RGBD images of 2 different views, have an adversarial loss on the RGB image, have a content loss between RGB1 and warp(RGB2), have a depth loss between D1 and warp(D2)
Equation 5:
- Possibly either "c_{1->2}" needs to be replaced by "c_{2->1}", or "G_{RGB}(z, c_1) - warp(G_{RGB}(z, c_2), c_{1->2})" needs to be replaced by "warp(G_{RGB}(z, c_1), c_{1->2}) - G_{RGB}(z, c_2)" (or am I missing something?)
- Not entirely sure why there is a different "projection" operation, since both "warp" and "projection" are calculated from Equation 3. I understand that "warp" is the combined Rt matrix that is estimated using the two views and Equation 3, assuming that the "d"s are correct. Not sure what "projection" does though, possibly explain it better?

DECISION: Very clearly written paper, simple idea executed well

The paper is clearly written and well organized. It uses a simple idea, and performs sufficient number of experiments to explore the idea. It is not very novel, but the paper shows its applicability with multiple architectures as a bonus.

The figures showed results almost only from their method. It would be great to pick one generator architecture, and elucidate more on the differences between not using their 3D loss and using it. Good attempt though.

ADDITIONAL FEEDBACK:
- Might not be "representation learning", instead it is learning a generative model.
- "3 EXPERIMETNS" -> "3 EXPERIMENTS"
- The appendix should have more details on the equations and the specific formulations of warp  and projection operations

**Experience Assessment:**

I have published one or two papers in this area.

**Review Assessment: Checking Correctness Of Derivations And Theory:**

I assessed the sensibility of the derivations and theory.

**Review Assessment: Checking Correctness Of Experiments:**

I carefully checked the experiments.

**Review Assessment: Thoroughness In Paper Reading:**

I read the paper thoroughly.

---

> ### Author Response · Authors · 2019-11-13
> **Response to Reviewer #3**
>
> We would like to thank the reviewer for valuable comments.
>
> Equation 5:
> - Figure 3 in [1] shows the detailed illustration of the "warp" operation. First, we calculate the position of pixels in G_{RGB}(z, c_1) when they are viewed from c_2 (c_{1->2} is used here), and warp G_{RGB}(z, c_2) according to the calculated positions with bilinear interpolation. Therefore, the relative transformation matrix we need is c_{1->2}. We will add more explanation in the paper.
> - Since the depth values in warp(G_{D}(z, c_2), c_{1->2}) are sampled from the depth values viewed from c_2, to compare G_{D}(z, c_1) with warp(G_{D}(z, c_2), c_{1->2}), we need to project the depth values of each pixel in G_{D}(z, c_1) to the viewpoint c_2. This is what we call "projection". We will add the explanation in the paper.
>
> Differences between not using 3D loss and using it
> - I will pick one generator and compare the generative results qualitatively in the paper. Because PGGAN and StyleGAN without 3D loss cannot control camera poses, we can only compare the random generation results. Comparisons on the 3D-latent-feature-based methods are already provided in Figure 4, 5 and Table 1, but we will add more results in the appendix.
>
> Representation learning
> - Learning generative models is often called "representation learning" because it can learn latent representations of the images [2, 3, 4, ...]. We will add a discussion about this in the paper.
>
> [1] Tinghui Zhou, Matthew Brown, Noah Snavely, and David G Lowe. Unsupervised learning of depth and ego-motion from video. In CVPR, 2017.
> [2] Alec Radford, Luke Metz, and Soumith Chintala. Unsupervised representation learning with deep convolutional generative adversarial networks. In ICLR, 2016.
> [3] Xi Chen, Yan Duan, Rein Houthooft, John Schulman, Ilya Sutskever, and Pieter Abbeel. Infogan: Interpretable representation learning by information maximizing generative adversarial nets. In NIPS, 2016.
> [4] Thu Nguyen-Phuoc, Chuan Li, Lucas Theis, Christian Richardt, and Yong-Liang Yang. Hologan: Unsupervised learning of 3d representations from natural images. 2019.

---

### Author Response · Authors · 2019-11-15
**Paper modifications**

We would like to thank all the reviewers for their valuable comments.

We revised the paper. The major modifications are as follows.
- Normalize depth images and visualize them with colormaps
- Separate the related works section from the introduction section
- Add some related works and discussions
- Additional results in the appendix
- Add results for point cloud visualization
- Quantitative evaluation on color and depth
- Add the explanation for "warp" and "projection" in the appendix
- Add the explanation for initial K in the appendix

---

### Decision · Program_Chairs · 2019-12-19

**Decision:**

Accept (Poster)

**Comment:**

The paper has initially received mixed reviews, with two reviewers being weakly positive and one being negative. Following the author's revision, however, the negative reviewer was satisfied with the changes, and one of the positive reviewers increased the score as well.

In general, the reviewers agree that the paper contains a simple and well-executed idea for recovering geometry in unsupervised way with generative modeling from a collection of 2D images, even though the results are a bit underwhelming. The authors are encouraged to expand the related work section in the revision and to follow our suggestion of the reviewers.